# The Challenges and Advantages of Distributed Fiber Optic Strain Monitoring in and on the Cementitious Matrix of Concrete Beams

**DOI:** 10.3390/s23239477

**Published:** 2023-11-28

**Authors:** Martin Weisbrich, Dennis Messerer, Klaus Holschemacher

**Affiliations:** Structural Concrete Institute, Leipzig University of Applied Sciences (HTWK Leipzig), 04275 Leipzig, Germany; dennis.messerer@htwk-leipzig.de (D.M.); klaus.holschemacher@htwk-leipzig.de (K.H.)

**Keywords:** distributed fiber optic strain monitoring, structural health monitoring, cementitious matrix, concrete beams

## Abstract

Distributed fiber optic strain measurement techniques have become increasingly important
in recent years, especially in the field of structural health monitoring of reinforced concrete structures.
Numerous publications show the various monitoring possibilities from bridges to special heavy
structures. The present study is intended to demonstrate the possibilities, but also the challenges,
of distributed fiber optic strain measurement in reinforced concrete structures. For this purpose,
concrete beams for 3-point bending tests were equipped with optical fibers on the reinforcement and
concrete surface as well as in the concrete matrix in order to record the strains in the compression
and tension zone. In parallel, an analytical approach based on the maximum strains in the uncracked
and cracked states was performed using the Eurocode 2 interpolation coefficient. In principle, the
structural design correlates with the measured values, but the strains are underestimated, especially
in the cracked zone. During load increase, structural distortions in the compression zone affected the
strain signal, making reliable evaluation in this zone difficult. The information content of distributed
fiber optic strain measurement in reinforced concrete structures can offer tremendous opportunities.
Future research should consider all aspects of the bond, sensor selection and positioning. In addition,
there is a lack of information on the long-term stability of the joint and the fiber coating, as well as
the effects of dynamic loading.

## 1. Introduction

Distributed fiber optic sensor systems (DFOS) offer interesting possibilities for strain and temperature measurement, especially in concrete construction [1,2,3,4,5]. There are already isolated applications in the field of Structural Health Monitoring (SHM) of structural and civil engineering, geotechnics or in special heavy construction [6,7,8,9,10,11,12,13,14,15,16,17]. These measurement systems have enormous potential, particularly in terms of sustainable long-term use of structures, but also in terms of improving civil safety by monitoring the structural health [18,19].

So far, the two measurement methods based on Rayleigh and Brillouin scattering have proven their suitability for measuring strain, cracks and crack development as well as temperatures along an optical fiber [20,21,22,23,24,25]. The measurement methods differ with respect to spatial resolution and maximum measurement length: while Brillouin scattering can achieve measurement lengths of up to 50 km to 80 km, Rayleigh scattering has a maximum measurement range of 50 m to 100 m [26,27]. Depending on the measurement length, spatial resolutions—i.e., the distance between the measurement points—from 1500 mm to < 1 mm can be realized [28,29,30,31]. Table 1 shows a basic comparison of the relevant characteristics of both methods.

Distributed fiber optic measurement offers advantages over point-based methods (e.g., strain gauges (STG), fiber Bragg gratings (FBG), or displacement transducers (DT) of all types). First, any area of the sensor can be used for measurement, allowing strain and temperature curves to be displayed over the entire measurement length. Second, fiber optic sensors can be bonded to the reinforcement or concrete surface [32,33], as well as embedded into the concrete matrix [16,34,35]. Table 2 lists other comparative values.

For a variety of SHM applications, the quality of the measurement signals is critical. The authors have validated fiber optic sensors in different scenarios within concrete construction. These include application to the reinforcement and concrete surface as well as integration into the concrete matrix [16,28,36,37]. Considering these experiences and further studies from the literature, strain transfer can be regarded as one of the major challenges [28,38,39,40,41,42], particularly when optical fibers protected by a coating or cable structure are used as sensors for DFOS (Figure 1). Since strain changes are only sensed in the fiber core [43], strain transfer losses may occur depending on the thickness and material of the coating or cable structure. These losses characterize the strain reduction between the substrate and the fiber core caused by shear deformation or slip. Especially when measuring in concrete structures before and during the casting process, robust sensor cables are often required to protect the sensitive fiber core [44]. Transfer functions can be used to adapt the strain transfer loss to the particular measurement task. In addition, these functions support the evaluation and interpretation of measurement signals [45].

Another major challenge is the application of the sensors to the substrate: the strain transfer losses described above can be amplified by a poorly executed bond or the use of inappropriate adhesives [45]. Signal distortions caused by deficient bond or adhesives are difficult to detect after installation. Therefore, refs. [28,36] defined procedures and adhesives for steel and concrete surfaces. It has been shown that the strain transfer losses at the concrete surface can be significantly reduced by removal of the cement skin, exposure of the aggregate and priming with a thixotropic epoxy resin.

In addition to sensor selection, proper adhesive application, and joint preparation, the integration into the concrete matrix is a significant challenge. It is important to install the sensors so that they remain in place and are functional after concreting, consolidation and curing. With respect to the strain transmission losses presented above, it is critical to find a compromise between strain sensitivity and robustness of the fiber optic sensors [44,46].

This paper discusses the advantages and challenges of distributed fiber optic sensing for strain measurement in concrete structures. Concrete beams tested in three-point bending tests using Rayleigh backscatter for strain measurement are used as an example. The particular significance of this study lies in the simultaneous measurement of strains within the concrete matrix, concrete surfaces, and reinforcement. In addition, this study provides important contributions to the behavior of fiber optic sensors in the compression zone. For the first time, the measurement results were also validated against existing design models for strain prediction. The present study shows that, if properly handled, results comparable to the design model can be obtained. According to the authors, only limited attention has been paid in the literature to the formation of the adhesive interface between the optical fiber and the substrate surface under test. This interface is one of the most important aspects for accurate strain measurement. In this regard, the authors provide guidance on the formation of the bond between the rebar or concrete surface and the fiber optic sensor in order to obtain optimum measurement results.

## 2. Experimental Program

### 2.1. Experimental Design

To demonstrate the capabilities and challenges of DFOS, three identical reinforced concrete beams of 0.15 m width and height and 0.70 m length were tested in a three-point bending test. The strain measurement technique used was Rayleigh backscatter (Luna Inc., Roanoke, VA, USA, ODiSi-B [47]) with a resolution of 2.6 mm. The reinforcement consisted of two diameter 10 mm rebars. In this series of tests, no transversal reinforcement was installed. For each beam, three cylinders with a diameter of 0.1 m and a height of 0.2 m were produced and cured according to DIN EN 12390-1:2021-09 [48] and tested following DIN EN 12390-3:2019-10 [49] to determine the material properties for the structural design.

Each beam was equipped with three fiber sensors at different locations. Within the tensile zone, one sensor was placed on one of the two rebars. The other two fiber sensors were placed in the compression zone of the beam, one of them in the matrix and the other one on the surface of the concrete. The sensor placement is shown in Figure 2.

The concrete beams were tested in a three-point bending test according to [50]. Steel plates with a thickness of 18 mm and a width of 50 mm, were used as supports and load application. With an effective span of 0.60 m, the load was applied at the center of the beam. The test setup and further relevant information on the specimens is shown in Figure 2.

After casting, the concrete beams were stored for eight days under controlled climatic conditions of 20${}^{\circ }C$ and 65 relative humidity before testing. The test was performed at five load steps (Table 3). The maximum load of 60 kN is within the range of the calculated shear capacity.

Before starting the first load step, a preload (without unloading) of approximately 3 kN was applied to check the test setup and to ensure correct alignment. At each load step, the strain state was recorded by the fiber sensors over a period of 5 s, which allows for the minimization of system-related measurement fluctuations in retrospect.

### 2.2. Concrete Mixture, Material of Application, and Fiber Types

In the experimental investigations, a high-strength, self-compacting, fine-grained concrete matrix was used. The cement is a compound from Dyckerhoff [52,53]. Table 4 lists the constituents of the matrix.

The use of a high-strength, fine-grained concrete is intended to minimize the variation between individual samples. In comparison to matrices with a larger maximum grain size, the inhomogeneity of the composition at a maximum grain size of 2 mm is minimized. Furthermore, the blurring caused by changing surface structures when applying the sensor to the concrete surface is reduced as much as possible compared to concretes with larger maximum grain size. Finally, the mechanical properties of high-strength concretes are largely determined by the more homogeneous cement matrix. Table 5 shows the companion specimen force and stress values obtained in compression tests.

A low attentuation loss fiber was used as sensor. For the strain measurement application, the coating material and the coating thickness are crucial, as they directly influence the strain transfer [16,28,36,38,39,40,41]. The fiber coated with Ormocer^®^ (organically modified ceramic) was applied to both steel and concrete surfaces and integrated into the concrete matrix [54]. Previous studies have investigated the suitability of the coating material for strain measurement [16,28,36]. The strain coefficient has been calibrated by the in-house calibration facility. Table 6 lists the main characteristics of the sensor used.

The adhesive used for application to the steel and concrete surfaces was Micro-Measurements’ M-Bond 200 (Wendell, NC, USA) [55], including the intended primer. The cyanoacrylate adhesive has shown good strain transfer properties in preliminary tests [36]. For the primer on the concrete surface, the epoxy resin SikaDur 330 from Sika Schweiz AG was used [56]. Previous tests have shown that priming with epoxy resin significantly improves strain transfer compared to the unprimed surface [28,37].

### 2.3. Application and Integration of the Sensors

The preparation and execution of the bonded area are critical to the strain transfer from the substrate to the sensor. Therefore, these aspects will be discussed in detail below. Adhesive bonding is not required for embedding the sensor in the matrix. However, steps must be taken to ensure a secure connection between the concrete matrix and the sensor, as well as positional stability before, during and after concreting.

The bonding process on the reinforcement surface is based on [36]. It includes the preparation and pretreatment of the bonded area and the application. For pretreatment, the surface must first be cleaned of corrosion residues. After roughening with grit size 200 and 400, the surface was cleaned with compressed air. The sensor and the surface were chemically cleaned with isopropyl alcohol. After fixing the fiber, the primer was applied. As shown in Figure 3, care should be taken to ensure optimal adhesive application [36,57]. Figure 4 shows the applied fiber sensor on the rebar.

The procedure for applying the fiber to the concrete surface is similar to the one described in [37]. The cement skin is removed to expose the aggregate in a first step. After blowing off substrate residues and dust with compressed air, as well as chemical cleaning with isopropyl alcohol, an epoxy resin is applied as a primer (Section 2.2). Once the epoxy has cured, the surface is wiped clean and the fiber is mounted and glued to the concrete surface.

Embedding the fiber into the concrete matrix requires a suitable installation aid to ensure that the fiber remains at the intended location after concreting. During the experimental investigations, a clamping device was developed to fix the fiber in the desired position. In addition, the clamping device allows the sensors to be tensioned immediately after concreting, in order to correct the position if necessary. Before concreting, it is essential to clean the fiber with isopropyl alcohol, as release agents in particular can interfere with the anchorage to the concrete.

## 3. Prediction of the Deformation

A structural design of the reinforced concrete beams for the maximum strains in the center and the strain profile over the entire length was performed to evaluate the results of the fiber measurements. Compression tests using cylinders with a diameter of 10 mm (Table 5) were used to determine the strength of the concrete. High-strength concrete mixes, in particular, exhibit large variations in stiffness. They can show deviations in the range of ±20% to 30% from the secant modulus calculated according to [58]. Based on previous tests with this concrete, a Young’s modulus of 45,000 N/mm^2^ is assumed. The longitudinal reinforcement bars have a cross-sectional area of $A_{s}=157$ mm^2^ and an elastic modulus of 200,000 N/mm^2^, according to [59].

To calculate the strains in the respective plane of the three fiber sensors, a separate determination is required for the uncracked and the cracked state. For all three beams, cracking started between load steps two and three. For the first two load steps, the cross-sectional values for the uncracked state were determined according to [51] as follows:

With the reinforcement ratio $\rho_{I}$ in the uncracked state, the ratio of the elastic moduli of concrete and reinforcement $\alpha_{e}$, as well as the height *h* and the static effective height *d* (Figure 2)
(1)$$\alpha_{e}=\frac{E_{s}}{E_{c}}$$
(2)$$\rho_{I}=\frac{A_{s}}{b\;\cdot \;h}$$
can be used to calculate both $A_{I}$ and $B_{I}$:(3)$$A_{I}=\alpha_{e}\;\cdot \;\rho_{I}\;\cdot \;\left(\frac{d}{h}\right)$$
(4)$$B_{I}=\alpha_{e}\;\cdot \;\rho_{I}$$

Starting from the upper edge of the beam, $k_{x,I}$ determines the position of the neutral axis $x_{I}$ in the uncracked state.
(5)$$k_{x,I}=\frac{0.5+A_{I}}{1+B_{I}}$$
(6)$$x_{I}=k_{x,I}\;\cdot \;h$$

With $k_{I}$, the moment of inertia of the area in the uncracked state $I_{I}$ can be calculated as follows:(7)$$k_{I}=1+12\;\cdot \;{\left(0.5-k_{x,I}\right)}^{2}+12\;\cdot \;\alpha_{e}\;\cdot \;\rho_{I}\;\cdot \;{\left(\frac{d}{h}-k_{x,I}\right)}^{2}$$
(8)$$I_{I}=\frac{k_{I}\;\cdot \;b\;\cdot \;h^{3}}{12}$$

The method for the derivation of the geometric quantities according to [51] is modified for the cracked state of the load step three as shown below:(9)$$\rho_{II}=\frac{A_{s}}{b\;\cdot \;d}$$
(10)$$A_{II}=B_{II}=\alpha_{e}\;\cdot \;\rho_{II}$$
(11)$$k_{x,II}=-B_{II}+\sqrt{B_{II}^{2}+2\;\cdot \;A_{II}}$$
(12)$$x_{II}=k_{x,II}\;\cdot \;d$$
(13)$$k_{II}=4\;\cdot \;k_{x,II}^{3}+12\;\cdot \;\alpha_{e}\;\cdot \;\rho_{II}\;\cdot \;{\left(1-k_{x,II}\right)}^{2}$$
(14)$$I_{II}=k_{II}\;\cdot \;\frac{b\;\cdot \;d^{3}}{12}$$

Tensile and compressive strains $\varepsilon_{s}$ and $\varepsilon_{c}$ are calculated as a function of fiber position at distance from neutral axis in center of field, together with moment load $M_{y}$ from load increments (Table 3).
(15)$$M_{y,i}=\frac{F_{i}\;\cdot \;l}{4}$$
(16)$$\varepsilon_{s/c,I/II,i}=\frac{M_{y,i}}{E_{c}\;\cdot \;I_{I/II}\;\cdot \;z}\;\cdot \;10^{6}\;\mathrm{in}\;\mu \varepsilon$$ where

F. . . applied load per load step according to Table 3;

i. . . load step running index;

l. . . effective span according to Figure 2.

To determine the strain values over the entire beam length, the interpolation coefficient ζ is proposed in [60]. The interpolation between the limit values is usually performed according to Equation (Equation 17). In relation to the respective location and the pure strain in the uncracked and the cracked state (Equation (Equation 16)), the average strain $\varepsilon_{m}$ at location *n* and load step *i* can be calculated.
(17)$$\varepsilon_{m,n,i}=\zeta_{n,i}\;\cdot \;\varepsilon_{II,n,i}+\left(1-\zeta_{n,i}\right)\;\cdot \;\varepsilon_{I,n,i}$$

## 4. Results

After the samples were tested, the raw data from the fiber sensors was processed. The sensor in the matrix of the third beam failed and could not record any data. The results of the individual sensors were evaluated separately for each sample. In general, especially for the measurement on the reinforcement, a good agreement with the limits in the uncracked and the cracked state was found. For the measurements on the concrete surface and in the matrix, the load application caused a disturbance of the concrete structure, which directly affected the strain measurement.

### 4.1. Strain Measurement on the Reinforcement Bar

Figure 5 shows the strain curves of the three specimens for each load step. The measured length is limited to the reinforcing bars. The black marks represent the maximum strain of the structural design in the uncracked and the cracked state, the dashed line shows the progression based on the interpolation coefficients according to [60]. The solid lines show the results of the individual fiber sensors for the different load steps. The strain curves in Figure 5 show good agreement with the calculated values from Section 3, especially for the load steps before crack initiation (LS 1 and 2). Cracking started just before the third load step reached. This is evident from the peak strain values in the center of the sample. Compared to the calculated values, a correlation can be seen, but higher strains can be observed, especially in the crack zone. Due to the more pronounced crack distribution in load steps 4 and 5, there are deviations in the strain curve between the specimens.

### 4.2. Strain Measurement on the Concrete Surface

Figure 6 shows the strain curves of the fiber sensors applied to the concrete surfaces separately according to load steps. Since the adhesive joint was only applied up to the supports, the strain measurement does not extend over the entire length of the specimen. While specimen 3 shows good agreement with the maximum strain determined by calculation, specimens 1 and 2 exhibit structural defects at the load application in the center of the specimen (350 mm, Figure 7). This leads to increased oscillation and variation in this area. In particular, for specimen 1, the microstructural disturbance affects the middle section of the strain measurement. Furthermore, all three sensors show identical strain outside the center.

### 4.3. Strain Measurement in the Cementitious Matrix

The strain curves of the measurements in the concrete matrix are shown in Figure 8. In specimen 3, the fiber sensor failed before the test was performed, so no values are available. The curves in uncracked state (LS 1 and 2) show good agreement with the structural design. From LS 3, the measured values of specimen 1 show oscillations in the area of the load application, which distort the strain signal (Section 4.2). The results of specimen 2, on the other hand, correlate with the structural design at all load levels shown.

## 5. Discussion and Conclusions

The results of the experimental investigations generally show good agreement between the strain measurement with DFOS and the structural design, although the maximum strains in the crack are underestimated by both calculation methods. The great potential of this measurement method for strain measurement in concrete construction and SHM is particularly evident from the results on the steel reinforcement. However, the measurements on the concrete surface and in the concrete matrix within the compression zone of the concrete also show the suitability of the measurement system. It can be concluded that sufficiently accurate results can be obtained by recommending appropriate application rules and materials.

The inhomogeneity and imprecision of the concrete can be potential sources of deviation between the fiber measurement and the structural design. This includes all factors related to specimen preparation, such as variations in composition, consolidation, and curing of the concrete. In addition, the Young’s modulus of both the concrete and the reinforcing steel and the tensile bending strength was determined using approximations from standards and preliminary research results [52,53]. Similarly, the applied calculation model is an approximation that does not take into account strain peaks in the cracks. The measured results show good agreement, especially for the first two load steps, despite the material and design uncertainties mentioned before.

As with all strain measurement methods, the design of the measurement project and the placement of the sensors in relation to the stresses is a challenge. In addition, the causes of measurement losses, deviations and fluctuations are manifold. In summary, the following conclusions can be drawn from the experimental investigations:Measured values in the area of punctual load application must be viewed critically or excluded. Adjusting or re-positioning would be beneficial here.All aspects of the bonded joint (pre-treatment, bonding process, and coating if necessary) form the basis for accurate strain measurement and must be performed carefully and professionally.Based on the strain signal, it is difficult or impossible to detect measurement errors due to defective bonded joints. This should be taken into account when using any method that is based on bonding for displacement or deformation measurement, especially under dynamic loading and harsh environment.

Compared to established measurement methods, DFOS offers decisive advantages: optical fibers are dielectric and insensitive to electromagnetic fields. In contrast to point-based measurement methods such as strain gauges, displacement transducers or fiber Bragg grating sensors, any area of the measurement fiber can be used for measurement, so that strain and temperature curves can be mapped over the entire measurement length. The optical fiber can either be bonded to almost any component surface or integrated into the material matrix of the component. This allows the sensor to reliably detect both load conditions due to external loads and deformations due to shrinkage or swelling.

Furthermore, DFOS can significantly extend the strain measurement compared to point-based methods. The results of the present investigations indicate an enormous potential in SHM and concrete construction. In addition to the evaluation of the deformation of a component, the prediction and detection of cracks, as well as the monitoring of curing processes, is possible at any point of the measuring fiber. The range of possible applications is enormous: bridges and engineering structures, canals, pipelines and hydraulic structures, as well as roads, can be monitored in a more targeted manner. This would not only make it possible to use existing resources more safely and sustainably, but also to better assess damage in the event of disasters or accidents. In addition to the challenges mentioned above, future research should focus on protecting the sensors from mechanical influences. Although fiber optic cables that are significantly more robust than optical fibers protected only by a coating are already available, the flexibility in mounting the sensors, the cost advantage and the strain sensitivity decrease at the same time. Also efforts in the area of bonded joints should be given high priority. In particular, more research is needed on strain transfer and long-term stability against moisture or chemical attack. The behavior of the bonded joint and the fiber coating at load ranges above the yield strength or under dynamic effects is also still unclear.

## Figures and Tables

**Figure 1 sensors-23-09477-f001:**
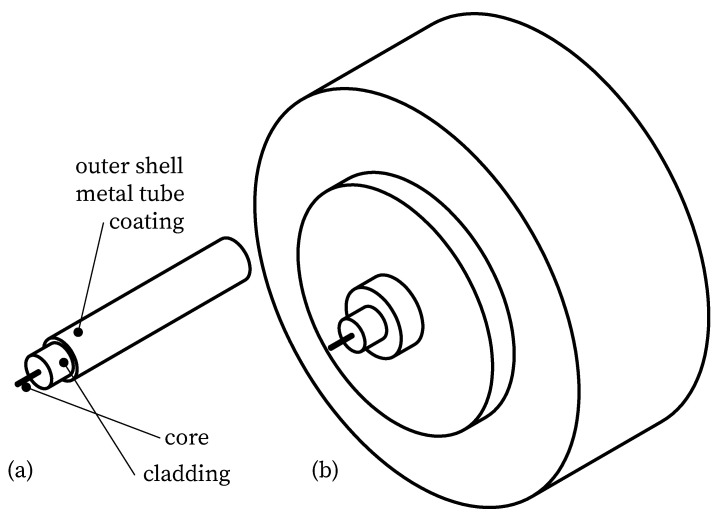
Exemplary scale structure of fiber optic sensors; (**a**) optical fiber protected only by coating; (**b**) structure of a sensor cable.

**Figure 2 sensors-23-09477-f002:**
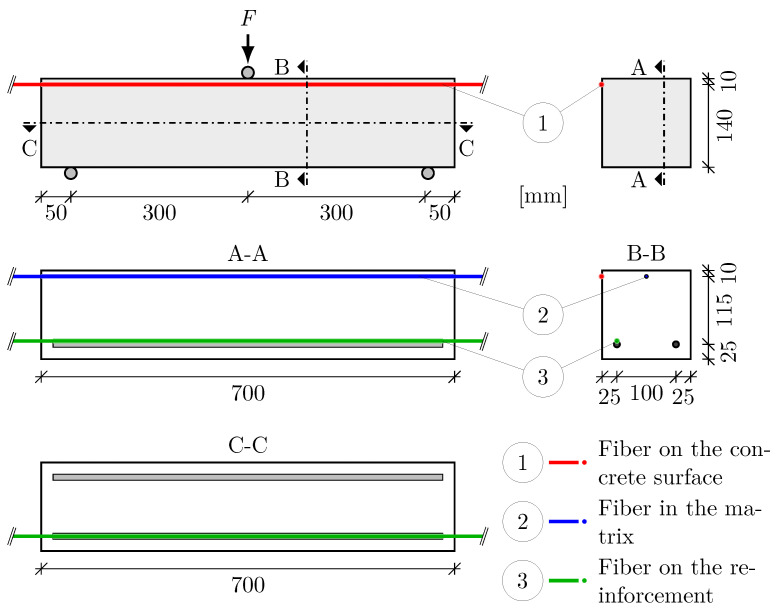
Experimental setup and sensor arrangement.

**Figure 3 sensors-23-09477-f003:**
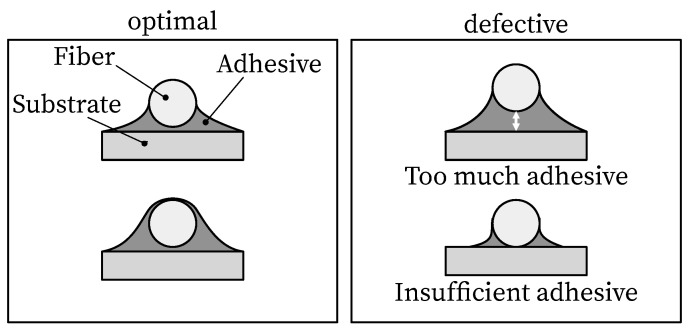
Optimal (**left**) and defective (**right**) adhesive joint according to [36,57].

**Figure 4 sensors-23-09477-f004:**
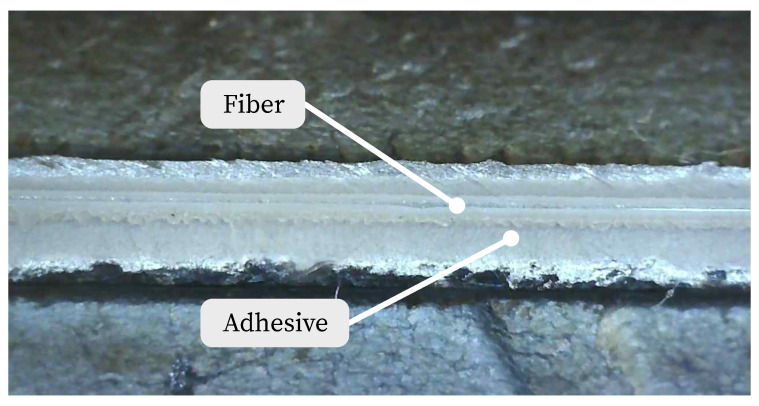
Fiber applied to the surface of the reinforcement.

**Figure 5 sensors-23-09477-f005:**
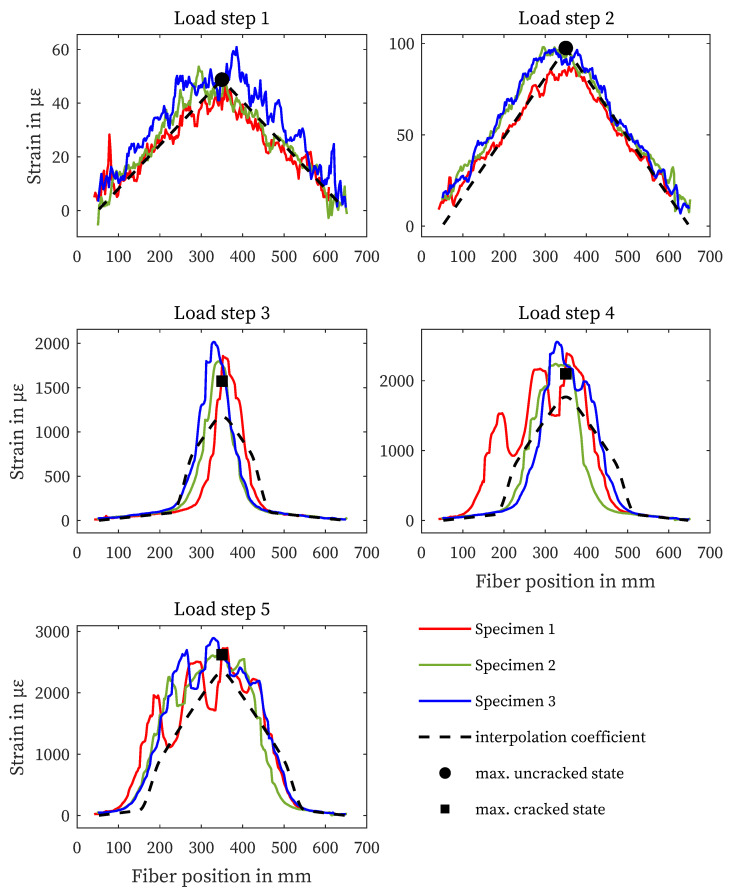
Results of structural design and rebar strain measurements for all specimens and load steps.

**Figure 6 sensors-23-09477-f006:**
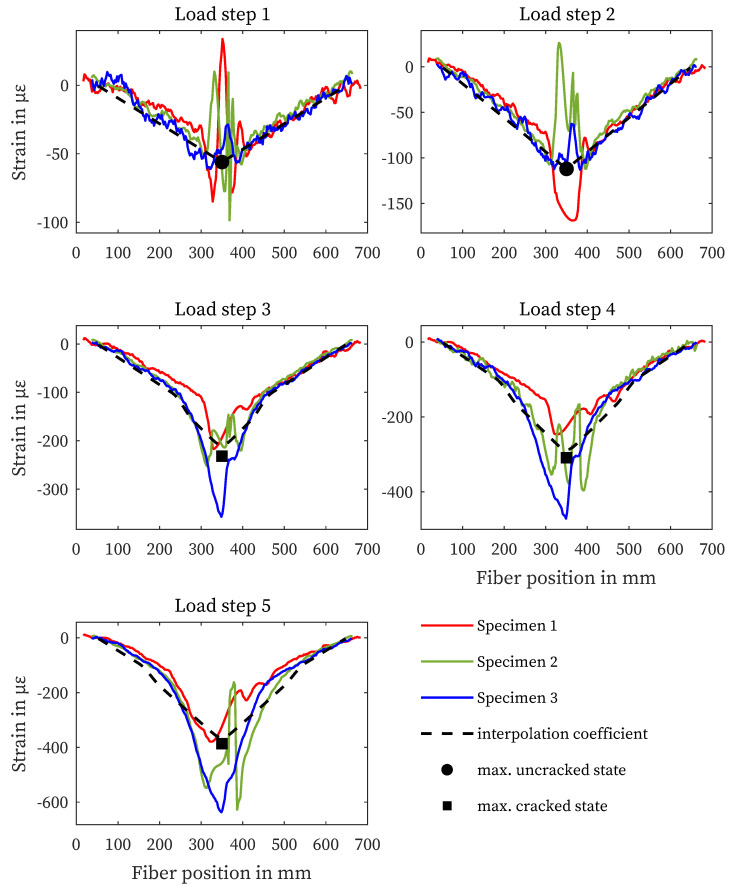
Results of structural design and concrete surface strain measurements for all specimens and load steps.

**Figure 7 sensors-23-09477-f007:**
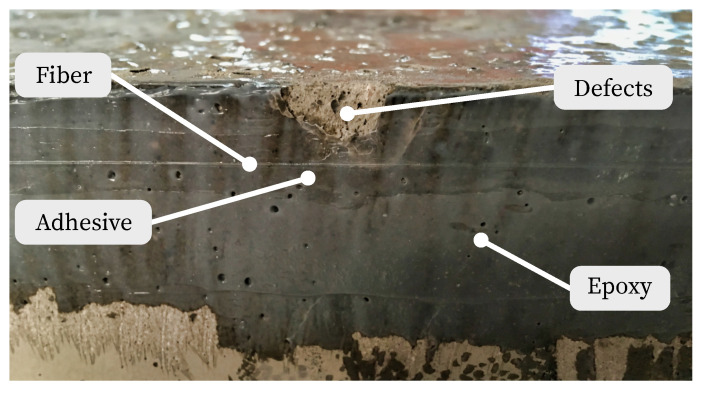
Applied fiber on the concrete surface and defects due to loading application.

**Figure 8 sensors-23-09477-f008:**
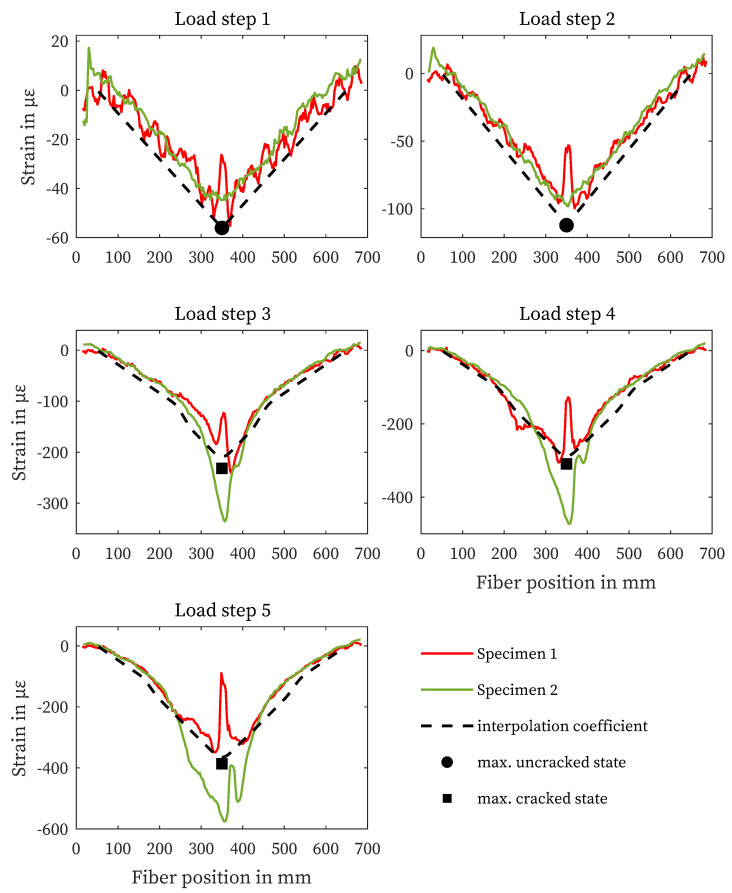
Structural design results and concrete matrix strain measurements for all specimens and load steps.

**Table 1 sensors-23-09477-t001:** Comparison of basic characteristics of measurement methods according to [26,27].

Characteristic	Rayleigh Scatter	Brillouin Scatter
Max. measuring range in m	50 to 100	80,000
Measuring rate in Hz	250	0.05 to 0.001
Resolution in mm	1.3 to 5.2	200 to 2500
Max. strain in με	15,000	30,000
Max. temperature in ${}^{\circ }C\;$	−40to200	−200 to 1000

**Table 2 sensors-23-09477-t002:** Comparison of displacement and strain sensors according to [28]. + positive/favorable; − negative/expensive; o neutral.

Method	STG	DT	FBG	DFOS
Electromagnetic influence	−	−	+	+
Resolution	−	−	o	+
Price per sensor	o	−	−	+
Price of measurement system	+	+	o	−
Amount of data	+	+	+	−
Information content per sensor	−	−	o	+
Combined temperature sensing	−	−	+	+
Measuring range	−	−	o	+

**Table 3 sensors-23-09477-t003:** Load steps, uncracked/cracked condition, moment, and calculated limit values from the structural design according to [51].

Load Step	Uncracked (I) Cracked (II)	Force	Moment	Strain Fiber pos. 1 and 2	Strain Fiber pos. 3
		kN	kN m	με	
1	I	12	1.8	−61	49
2	I	24	3.6	−121	98
3	II	36	5.4	−310	1573
4	II	48	7.2	−414	2097
5	II	60	9.0	−517	2621

**Table 4 sensors-23-09477-t004:** Concrete composition in $kg/m^{3}\;a$ccording to [52,53].

Matrix	Quantity
BMK-D5-1 (Compound)	815
Sand BCS 0.06/0.2	340
Sand 0/2	965
Water	190
Superplasticizer (MC-VP-16-0205-02)	17

**Table 5 sensors-23-09477-t005:** Concrete cylinder test results.

	Specimen	Ultimate Force in kN	Compressive Strength in N/mm^2^
beam 1	1.1	898	114
	1.2	879	112
	1.3	896	114
beam 2	2.1	832	106
	2.2	890	113
	2.3	867	110
beam 3	3.1	874	111
	3.2	888	113
	3.3	861	110
	Mean	876	111

**Table 6 sensors-23-09477-t006:** Fiber sensor characteristics.

Description	Ormocer ^®^
Fiber type	LAL-1550-125
∅ Core in µm	9
∅ Cladding in µm	125(1)
∅ Coating in µm	195
Attenuation in dB/km	<2.5
Strain coefficients in με/GHz	−6.67

## Data Availability

The datasets used and analyzed in the current study are available from the corresponding author on reasonable request.

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
