# Peer review of "The Challenges and Advantages of Distributed Fiber Optic Strain Monitoring in and on the Cementitious Matrix of Concrete Beams"

_sensors, 2023, doi:10.3390/s23239477_

Round 1
Reviewer 1 Report
Comments and Suggestions for Authors
In this manuscript, three concrete beams for 3-point bending tests were equipped with optical fibers on the reinforcement and concrete surface as well as in the concrete matrix in order to record the strains in the compression and tension zone. Based on the detailed experimental data, the challenges and advantages of distributed fiber optic strain monitoring in and on the cementitious matrix of concrete beams are investigated. The manuscript was prepared with high quality, so I would like to recommend it to be accepted for publication. The following are some minor revision suggestions:
1. What technique is used for the strain measurement in the experiments? Please give the details of the demodulation technique or instrument product information.
2. In Table 6, the units for core, cladding and coating size are not correct.
3. The sub-figures in Fig. 5, 6 and 8 should be numbered and introduced in their captions respectively.
Comments on the Quality of English LanguageCan be improved further.
Author Response
Dear Reviewer,
Thank you for your comments. I have added the following text in response to your comments.
Line 70: Information about the measurement technique
Line 86 and reference [47]: Information about the measurement technique
Table 6: correct unit
Figure 5, 6, 8: Caption
Best regards,
Martin Weisbrich

Reviewer 2 Report
Comments and Suggestions for Authors
The manuscript conducted an interesting experimental work regarding the use of DFOS for the measurement of strains in concrete structures. The experiment was well designed and some illuminating discussions were presented. The reviewer would like to seek the authors opinions on the durability of DFOS in real-life scenarios. Would it survive long-term harsh environment, repetitive fatigue load, etc.? This is important for its actual applications.
Author Response
Dear Reviewer,
Thank you for your comments. I have added the following text in response to your comments.
Line 258-261: Conclusion about dynamic loads and harsh environment
Best regards,
Martin Weisbrich

Reviewer 3 Report
Comments and Suggestions for Authors
It seems to me that there are some inaccuracies in Table 2. For example, it says that strain gauges (STG) are not affected by electromagnetic radiation, while FBG and DFOS are affected by electromagnetic radiation. This seems wrong to me. Unfortunately I could not find the paper [28] Weisbrich, M. Verbesserte Dehnungsmessung im Betonbau durch verteilte faseroptische Sensorik. phdthesis, Technische 361 Universität Bergakademie Freiberg, 2020. to which Table 2 refers. I suggest that the authors check the whole Table 2 carefully. Especially since in row 260 the authors contradict Table 2.
I found Figure 1,a in the paper https://doi.org/10.1002/best.202100057, but there is no reference to it under the figure. I found the same illustration in other works as well.
The description of a particular cement grade and composition would seem to me to be more appropriate, for example, in the journal materials or applied science. In the journal Sensors, it would be more reasonable to put more emphasis on the sensors themselves. Tables 4, 5, 6 would also be more appropriate in a paper with applied research rather than in Sensors, where the main emphasis is on the sensors themselves and their applications rather than on analyzing the response of individual material types.
In the section "2.3. Application and integration of the sensors" the authors pay much attention to the optimal and defective bonding of the fiber to the reinforcement. However, they did not specify the optimum thickness and grade of the adhesive. And they have not given the dependence of the optimum adhesive thickness on the surface and adhesive grade. This is especially surprising compared to the way the composition of concrete was described in detail.
On line 170, the sentence is incomplete, it has a ":" at the end and needs to be continued.
I did not find the height h and the static effective height d 172 in Figure 2, see line 172.
I was confused by the coefficient 12 in two places, in equation (7) and in equation (8). It would be nice if the authors would cite the source.
Unfortunately, after first reading the article, I didn't catch what method the authors used to measure. I did not find an indication of the measurement conversion method. Probably this should have been emphasized separately, and an optoelectronics measurement scheme should have been given.
In the manuscript, I saw rather a comparison of the modeling method with experimental data. But there is no specific description of how the original experimental data was obtained and what measurement conversion method was used.
The inputs are not suitable for a logbook. They only state that the modeling agrees with the experiment.
The last paragraph of the conclusions is generally appropriate for any fiber optic measurements.
I would encourage the authors to submit the manuscript to another journal or seriously revise it.
Author Response
Dear Reviewer,
Thank you for your comments. I have made the following improvements based on your comments.
Figure 1: Replaced and new caption
Line 70: Information about the measurement technique
Line 86: Information about the measurement technique
Source 28 can be found in full text in any search engine.
Table 2 shows exactly what it should, especially in combination with the statement in line 260.
In fiber optics, the internal laser is isolated from external light sources. High frequency EMI (i.e., stray or ambient light) from outside hitting a bare optical fiber can't travel down the fiber in a stable manner because it hits it at the wrong angle, and if the optical fiber is encased in an opaque material, external light can't make contact with the fiber at all.
Lower frequency EMI, such as radio waves, can penetrate the coating and enter the fiber. The typical way EMI creates noise is by inducing charge movement in a conductor, but the fiber is not a conductor, so no charge movement can be induced. And even if it were a conductor where charge motion could be induced, it would not matter because the optical fiber carries its signal with light, not charge motion. Therefore, optical fiber is immune.
For example, in https://doi.org/10.3390/s23218933, https://doi.org/10.3390/s23218869 or https://doi.org/10.3390/s23218775 or in many other sensor papers, information about the concrete used is presented. Furthermore, this information is relevant to the interpretation of the results.
Regarding the thickness of the adhesive layer, there is no precise information in the literature. However, the fiber diameter is known, so an approximation can be made.
The sentence in line 170 is not incomplete, the following explanation is implied.
Figure 2 shows d and h.
References for all formulas are given in the text. More than enough references are given for the conversion of measured values, and in my opinion they do not need to be described in detail in every paper.
Both in the evaluation and in the conclusion it is pointed out that the measurement results do not completely agree with the model values.
I don't understand the comment about the log book.
Best regards,
Martin Weisbrich
